# Basic Needs in Horses?—A Literature Review

**DOI:** 10.3390/ani11061798

**Published:** 2021-06-16

**Authors:** Konstanze Krueger, Laureen Esch, Kate Farmer, Isabell Marr

**Affiliations:** 1Department of Equine Economics, Faculty of Agriculture, Economics and Management, Nuertingen-Geislingen University, Neckarsteige 6-10, 72622 Nürtingen, Germany; laureen.esch@gmx.de (L.E.); ISY-MARR@web.de (I.M.); 2Zoology/Evolutionary Biology, University of Regensburg, Universitätsstraße 31, 93053 Regensburg, Germany; 3Department of Animal Welfare, Ethology, Animal Hygiene and Animal Husbandry, Ludwig Maximilian University Munich, Veterinarian Medicine, Veterinaerstr. 13/R, 80539 Munich, Germany; 4Centre for Social Learning & Cognitive Evolution, School of Psychology, University of St Andrews, St Andrews, Scotland KY16 9JPh, UK; katefarmer74@gmail.com; 5Behavioural Physiology of Farm Animals, University of Hohenheim, Garbenstr. 17, 70599 Hohenheim, Germany

**Keywords:** abnormal behaviour, active responses, horse, movement, passive responses, roughage, stress, social contact

## Abstract

**Simple Summary:**

All animals have requirements that are essential for their welfare, and when these basic needs are not met, the animal suffers. In horses, it is claimed that these needs include social contact, social companionship, free movement and access to roughage in the form of grass, hay and/or straw. To validate this claim, this review examines 38 studies that reported on horses’ responses when one or more of these factors are restricted. We categorised the type of responses investigated: (a) Stress (e.g., increased stress hormones), (b) Active (e.g., increased aggression), (c) Passive (e.g., depressive-like behaviour) and (d) Abnormal Behaviour (e.g., stereotypies), and analysed the frequencies with which the investigated responses were shown. Overall, the studies reported that horses did react to restrictions in the described basic needs, especially to combinations of restricted social contact, free movement and access to roughage. The observation of passive responses and the development of abnormal behaviour provided compelling evidence that horses were suffering under these restrictions, and existing abnormal behaviours indicated that they had suffered at some time in the past. We conclude that the literature supports the claim that social contact, free movement and access to roughage are basic needs in horses and need to be taken into consideration to ensure their mental and physical welfare in management and training.

**Abstract:**

Every animal species has particular environmental requirements that are essential for its welfare, and when these so-called “basic needs” are not fulfilled, the animals suffer. The basic needs of horses have been claimed to be social contact, social companionship, free movement and access to roughage. To assess whether horses suffer when one or more of the four proposed basic needs are restricted, we examined several studies (*n* = 38) that reported behavioural and physiological reactions to these restrictions. We assigned the studies according to the four types of responses investigated: (a) Stress, (b) Active, (c) Passive, and (d) Abnormal Behaviour. Furthermore, the number of studies indicating that horses reacted to the restrictions were compared with the number of studies reporting no reaction. The limited number of studies available on single management restrictions did not allow conclusions to be drawn on the effect of each restriction separately, especially in the case of social companionship. However, when combinations of social contact, free movement and access to roughage were restricted, many of the horses had developed responses consistent with suffering. Passive Responses, indicating acute suffering, and Abnormal Behaviour, indicating suffering currently or at some time in the past, were especially clearly demonstrated. This provides further evidence of the usefulness of assessing behavioural parameters in combination with physiological measurements when evaluating horse welfare. This meta-analysis of the literature confirms that it is justified to claim that social contact, free movement and access to roughage are basic needs in horses.

## 1. Introduction

Every animal species has particular environmental requirements that are essential for its welfare [1,2,3] and these are described as basic needs. As a general assumption, it has been claimed that social contact, social companionship, free movement and access to roughage are horses’ basic needs [4,5,6,7,8,9,10]. Horses are said to need social contact because in a natural setting they live in large groups, with about 200–400 horses comprising a herd. These herds are divided into subgroups of harems (usually composed of one to five males, several females and their offspring) and bachelor bands (composed of males of different ages) [4,5,7,9,10]. Furthermore, horses are assumed to need social companionship because about one third of all horses form stable social bonds with members of their subgroup. Bonded animals mutually protect each other and their offspring, as well as protecting resources such as food, water and resting places [4,5,9]. In addition, it has been claimed that horses need free movement because under natural conditions, they cover between 3 to 30 km daily [4,6,7,8]. Finally, horses have been said to need access to roughage as, in nature, they feed on grass for 12 to 16 h per day [4,5,7,8].

However, it must be evaluated whether keeping horses under human management conditions that restrict their basic needs compromises welfare. Therefore, several studies have set out to assess whether horses suffer and, if so, which responses demonstrate suffering, when one or more of these needs are not fulfilled.

Many of these studies analysed whether horses reacted to such restrictions with metabolic [11,12,13,14], physiological [15,16,17,18,19,20,21,22,23,24,25,26], behavioural and/or cognitive [10,25,26,27,28,29,30,31,32] signs of stress [11,12,13,14,15,16,17,18,19,20,21,22,23,24,25] that would indicate reduced welfare [27]. Horses may, for example, develop gastric ulcers caused by physiological stress and when access to roughage is restricted and gastric acidosis cannot be buffered by the feed and saliva amylase secreted during feeding [27]. Stress under restricted management conditions may also affect the animals’ emotional state and preferences for processing information in one or other brain hemisphere. [25,28,29,30,31,32]. Marr et al. [25], found that left shifts in horses’ motor and sensory laterality were useful behavioural indicators of changes in information processing in particular brain hemispheres when horses experienced stress from a change from group management to individual housing with initial training. Pioneering comparisons between hemispheric electroencephalogram (EEG) wave patterns in horses and those typical of emotional arousal in humans provide further evidence that horses respond negatively to restrictions in movement and social contact [26]. Furthermore, Löckener et al. [10] found that horses develop positive expectations towards their environment, i.e., a positive cognitive bias, when moved to group housing after experiencing management restriction.

Increased displays of certain behaviours in response to restrictions in basic needs have also been evaluated. Horses may show increased aggressive behaviour towards each other [18,22,33,34] and/or towards people [26,35], especially when their social relationships are disrupted [11,18,36,37,38]. Horses may show more interest in novel objects [38,39,40] and increased cooperativeness during training (i.e., trainability) when they are stabled in social groups rather than individually [13,21]. They may seek close proximity to their conspecifics more frequently when their social companionship is disrupted, as has been evaluated by applying nearest neighbour analysis [36]. Furthermore, horses may become more active when free movement, either individually or in social groups, is restricted [14,15,19,21,24,38,39], or when foals are weaned and separated from dams and social companions [40]. They may also show more hurried eating behaviour when roughage is limited [14]. On the other hand, horses may respond to restrictions in basic needs by reducing certain behaviour displays [11,15,18,19,20,21,22,24,26,35,38,39,40,41,42,43] and may even show depressive-type symptoms [22].

Behavioural disorders have also been assessed in horses [17,44], including stereotypies [8,26,45,46,47,48] and self-harming (i.e., redirected behaviour [8]). It has been debated whether abnormal behaviour is actually harmful or is rather a behavioural adaptation to a poor environment [47]. In this respect, most authors agree that stereotypies can be considered maladaptive behaviour indicating that horses are suffering under their housing or training conditions, and they give various reasons for this conclusion. Firstly, horses develop stereotypies in housing in which movement is reduced and there is little social contact. Secondly, free ranging, feral horses have never, to date, been observed showing stereotypic behaviour [7]. Thirdly, animals that displayed redirected behaviour and stereotypies also showed clinical signs of reduced welfare. These included: (a) self-inflicted skin lesions as a result of redirected behaviours, (b) gut damage, including lesions, ulceration and damaged mucosal tissue as a consequence of a wind-sucking and crib-biting [27,49] and (c) increased frequency of laminitis as a result of stereotypic movements such as weaving and box walking [27].

This literature review aims to provide an overview of studies that have evaluated the effects of changes in factors relating to horses’ environmental requirements (specifically social contact, social companionship, free movement, and access to roughage) on behavioural or physiological parameters (Table 1). We asked (a) whether a meta-analysis of the relevant literature supports the claim that social contact, social companionship, free movement and unlimited access to roughage are basic needs in horses, and (b) whether certain measurements can be considered reliable indicators for the analysis of animal welfare and basic need restrictions.

## 2. Materials and Methods

From August 2020 to February 2021, we searched the research platforms Research Gate, PubMed, Web of Science, Science Direct and Google Scholar for studies on social contact, social companionship, free movement and unlimited access to roughage in horses. We identified 38 studies (Table 1 and Appendix A) on behavioural and physiological responses to management conditions in which one or more of the four proposed basic needs were restricted, and this will be the basis of this literature review. Information on the horses observed in the studies and the management conditions they lived under are given in the Appendix A. Of the studies we identified, 17 evaluated horses’ responses to the given management situation, and 21 studied horses’ responses to changes in management conditions. One study is cited, but not included in the analysis, as it describes the horses’ responses to particular management conditions but does not analyse responses to management restrictions.

As horse housing is a complex setting, isolating only one of the aspects is very difficult. Therefore, only a few studies have analysed changes in only one of the proposed basic needs, and most consider two or more needs simultaneously (see Table 1). These generally compared two or more horse groups under housing conditions that differed in one or more needs (see Table 1), but a few examined changes in housing conditions of one group.

### 2.1. Data Processing

Four steps were applied in the evaluation of the literature (Table 1 and Figure 1). Firstly, the literature was assigned to the proposed basic needs that were restricted. Secondly, the studies were categorized according to the type of response shown by the horse, and thirdly, we assessed whether changes in the behavioural and physiological reactions indicated that the restrictions were compromising the horses’ welfare. Finally, the frequencies of studies reporting responses were compared with the frequency of those reporting no responses to the restrictions.

The studies were grouped according to the proposed basic needs that were restricted (Table 1):No basic need restricted, *n* = 1Social Contact restricted, *n* = 4Social Companionship restricted, *n* = 1Free Movement restricted, *n* = 4Access to Roughage restricted, *n* = 3Social Contact and Access to Roughage restricted, *n* = 1Free Movement and Social Contact restricted, *n* = 16Free Movement and Access to Roughage restricted, *n* = 1Free Movement, Social Contact and Access to Roughage restricted, *n* = 7

The studies were grouped according to evaluated behavioural and physiological measurements. The terms “shown” and “not shown” indicates whether the horses displayed any of the following types of responses to management conditions or not (Table 1, Figure 1):

a. Manuscripts evaluating ‘Stress Responses’, *n* = 16, methods of measuring ‘Stress Responses’, *n* = 21

i. shown, *n* = 14

ii. not shown, *n* = 7

Stress responses include increased metabolic rates and reduced body condition score [11], reduced growth rate [12], reduced eye temperature [13] and triiodothyronine and thyroxine excretion [14]. Physiological responses include changes in cardiovascular functions (heart rate [15,16,17,18,19], heart rate variability [19,20,21]), excretion of catecholamines [14], changes in stress hormone levels (blood cortisol [14,15,22], salivary cortisol [16,19,23], faecal glucocorticoid metabolites [12,13,17,20,24,25]), faecal immunoglobulin A [25], increased gastric acidosis [27] and changes in EEG wave patterns [26]. Behavioural parameters comprise changes in horses’ motor and sensory laterality [25,28,29,30,31,32] and changes in their positive or negative expectations towards their environment, i.e., in their cognitive bias [10].

b. Manuscripts evaluating ‘Active Responses’, *n* = 24, methods of measuring ‘Active Responses’, *n* = 33

i. shown, *n* = 22

ii. not shown, *n* = 11

Horses may show increased aggressive behaviour towards each other and/or towards people [11,18,22,26,33,34,35,36,37,38]. They may show more interest in novel objects [38,39,40] and increased cooperativeness during training (i.e., trainability) [13,21]. They may seek close proximity to their conspecifics more frequently, as observed by applying nearest neighbour analysis [36]. Furthermore, horses may become more active [14,15,19,21,24,38,39,40] and may also show more hurried eating behaviour [14].

c. Manuscripts evaluating ‘Passive Responses’, *n* = 17, methods of measuring ‘Passive Responses’, *n* = 23

i. shown, *n* = 15

ii. not shown, *n* = 8

Passive responses include reduced reactivity towards stimuli and human presence [15,22,26,40,41], reduced close contact with conspecifics [35] and reduced trainability [15,35,43]. Reduced activity [42] was evaluated by measuring the time spent lying down [11,21,39], moving [11,19], and the distance moved [20]. Some horses may show depressive-like behaviours [22].

d. Manuscripts evaluating ‘Abnormal Behaviour’, *n* = 14, methods of measuring ‘Abnormal Behaviour’, *n* = 17

i. shown, *n* = 14

ii. not shown, *n* = 3

Abnormal Behaviours include self-harming (i.e., redirected behaviour such as self-biting [8]), stereotypies such as crib-biting, wind-sucking, weaving and box-walking [14,41,48], as well as other behaviours such as, wood-chewing, bed-eating, manure-eating, rug-chewing or tearing, stable kicking, aggression towards humans and masturbation [8,44,55].

### 2.2. Data Analysis

The R-Project statistical software (R Development Core Team 2021, https://www.r-project.org/) was used for the statistical analysis and Excel for creating the figures when analysing the frequencies of studies on basic need restrictions. Some of the data were not normally distributed (Shapiro–Wilk Test). Therefore, Generalized Linear Models (GLMs) for multivariate testing with fixed factors were applied. For a general approach, we analysed the frequency of studies reporting reactions or no reaction under the particular restrictions of basic needs and the different responses shown. The GLM (formula = number of manuscripts evaluating response~response versus no response + response type, family = poisson (identity), data = Dataset) was used. We continued by applying a nested Generalized Linear Model (GLM) to analyse whether manuscripts reported differences for showing or not showing responses nested within the different types of response. Therefore, the GLM (formula = number of manuscripts. evaluating response~response versus no response % in % response type, family = poisson (identity), data = Dataset) was applied. The full statistical data are given in the Appendix A. Binomial Tests were applied to compare the frequencies of showing or not showing certain types of responses under certain management restrictions. All tests were two-tailed, and the significance level was set at 0.05.

## 3. Results

Overall, a significant number of studies reported that horses did show responses to restrictions in the proposed basic needs (responses shown: *n* = 90, not shown: *n* = 29; GLM: *n* = 37, z = 4.08, *p* < 0.001; Table 1, Figure 1, Appendix A). However, reports on whether horses showed responses differed between the particular response types (GLM: *n* = 37, z = −2.57, *p* = 0.01; Figure 1, Table 1, Appendix A).

Of the studies on changes in behavioural and physiological measurements of ‘Stress Responses’, *n* = 13 studies reported that horses showed responses and *n* = 7 did not (Table 1, Figure 1). The studies did not clearly indicate stress responses for single restrictions, as when studies found several stress measurements changed, i.e., indicating stress, the same study or other studies also found other stress measurements remained unchanged, i.e., indicating no stress. When the reactions to all the different restrictions were compared, the number of reports showing stress responses correlated with the number of reports showing no response (GLM: *n* = 37, z = 2.49, *p* = 0.01; Figure 1, Table 1, Appendix A).

The same was true for the ‘Active Responses’, with *n* = 14 studies reporting that active responses were shown and *n* = 8 reporting they were not (Figure 1, Table 1). Again, while many parameters changed almost as many did not change when analysing a particular restriction and there was a general correlation between active responses and no active response over all the basic need restrictions (GLM: *n* = 37, z = 2.63; *p* = 0.008, Figure 1, Table 1, Appendix A).

‘Passive Responses’ were reported in *n* = 14 studies and no ‘Passive Responses’ for *n* = 8 studies (Figure 1, Table 1). ‘Passive Responses’ to particular restrictions indicated more clearly than ‘Stress Responses’ and ‘Active Responses’ that horses responded with changes in behaviour to certain restrictions. The literature revealed only a trend in correlation between the horses showing passive responses to those showing no passive response when comparing all the different restrictions in the basic needs (GLM: *n* = 37, z = 1.85, *p* = 0.06; Figure 1, Table 1, Appendix A).

Finally, a significant number of horses demonstrated the response ‘Abnormal Behaviour’ in response to any restriction in basic needs (‘Abnormal Behaviour’: *n* = 12, no ‘Abnormal Behaviour’: *n* =3, Binomial test: *p* = 0.02, Figure 1). Horses reacted more clearly by either showing abnormal behaviour or not showing abnormal behaviour to particular restrictions, as there was again only a trend for a correlation between reports of abnormal behaviour and reports of no abnormal behaviour across the studies on all the different restrictions examined (GLM: *n* = 37, z = 1.74, *p* = 0.08; Figure 1, Table 1, Appendix A).

The literature provides no clear conclusion as to whether horses generally show responses to restrictions in just one of the parameters of social contact, social companionship, free movement and access to roughage in isolation (Figure 1). For most of the comparisons the sample sizes were too small for applying statistical tests, the few that allowed statistical testing revealed no difference between measurements that show and those that do not show a response to the restrictions (Binomial tests: all *p* > 0.05; Figure 1). When social contact, free movement and access to roughage were mutually restricted, horses showed behavioural and/or physiological responses (Binomial test: *p* < 0.001; Figure 1).

## 4. Discussion

The aim of this study was to evaluate whether certain measurements can be considered reliable indicators for the analysis of animal welfare under basic need restrictions by analysing the results published in the literature. The interpretation of the studies calls for caution as positive reporting biases may result in more studies that prove a certain measurement to be effective than those reporting no effect [56]. However, the literature also contains several studies that actually report missing responses of horses to the measurements under observation. We therefore found the results of the meta analysis of these studies worth discussing.

Especially the development of abnormal behaviour and stress responses under the long-term stress of compromised environmental conditions has been considered a maladaptive strategy which does not provide the animals with options to deal with such conditions [8,57,58]. When animals develop these responses, they are said to be clearly suffering [58,59]. The present literature review provides evidence that this claim is justified in horses, especially in relation to abnormal behaviour. Horses develop significant levels of abnormal behaviour when social contact, social companionship, free movement and access to roughage are compromised. Therefore, it appears to be justified to use the display of abnormal behaviour as a behavioural animal welfare indicator when analysing the quality of horse housing and training, as has been established for animal welfare protocols [60].

However, it remains debatable whether a long-term display of abnormal behaviour provides strong evidence of a horse suffering under its present management conditions. Stereotypic behaviour may have developed under previous conditions and persist, even when management and training return to favourable conditions [45,46,57]. A recent development in EEG wave pattern analysis [26] offers promising new insights into this debate. Horses that had consistently displayed stereotypic behaviour for at least one year and lived under restricted management conditions showed EEG wave patterns comparable to those indicating negative emotional states in humans. However, caution should be exercised when evaluating welfare in horses displaying stereotypic behaviour as horses secrete dopamine when engaging in such behaviour and this elicits a positive emotional state in the animal (see for review: [61]).

Similarly, the measurement of behavioural and physiological stress parameters did not necessarily indicate that all horses experienced stress in compromised management conditions (Figure 1). Some studies found that horses did respond with behavioural or physiological stress parameters, and others did not (Table 1). This discrepancy may be due to difficulties in assessing long-term stress. When animals suffer stress for longer periods, such as when their environmental requirements are restricted for a long time, several physiological parameters, such as stress hormones and cardiovascular functions may return to base levels or below [13,25,62,63].

There are promising stress parameters that may allow long-term stress to be evaluated, such as immune cell suppression, changes in motor laterality [25,64], increased hemispheric laterality and EEG wave patterns analogous to those measured in humans with negative emotional arousal [26]. However, there is not yet sufficient evidence for these to be included in the present literature survey because they have only been studied in a few pioneering papers [25,26,64,65].

There may also be individual differences between horses in their stress resistance and the importance an individual attaches to any particular stressor [63]. In addition, previous experience with the restrictions in basic needs may be a factor. For example, horses that were born and raised in a stabled environment may be less stressed by restricted movement than horses raised at grass and then moved to a stabled environment [25]. Moreover, horses that had previously experienced individual housing did not display any significant differences in physiological stress responses between individual housing with semi-contact to conspecifics and group housing [13,24], whereas horses that were naïve to individual housing showed significant physiological stress responses when moved from group to individual housing [17,19,25]. A combination of several physiological and behavioural stress parameters may provide the strongest evidence for stress in horses as some studies found conflicting results when comparing a limited spectrum of physiological and behavioural data [63].

However, the proportion of horses showing passive responses supports the claim that many horses suffer long term stress under the investigated management restrictions [22,52,66]. Passive responses, such as reductions in activity, feeding, behaviour displays, contact to persons or other conspecifics and reactions to the environment indicate that horses withdraw from external stimuli and may show a depressive-like state [22,66]. These responses are maladaptive for animals such as horses, which are both flight animals that rely on fast responses to acute challenges and social animals that rely on fast responses to social challenges [9].

Interestingly, the analysis of ‘Active Responses’ did not provide a clear conclusion. Some studies reported that horses showed active responses when the animals were faced with restricted basic needs and others did not (Table 1 and Appendix A). It may be difficult to clearly distinguish the level of activity that constitutes a stress response, as elevated aggression and movement may counteract mild stress [34,37].

As only very few studies succeeded in isolating the effects of restrictions in just one of the four proposed basic needs of social contact, social companionship, free movement and access to roughage, a clear statement on whether animals can generally cope with the particular restrictions remains elusive. However, there were sufficient studies on combined restrictions in social contact, free movement and access to roughage, and when the horses’ environments were restricted in these three conditions, they appeared to suffer, as ‘Abnormal Behaviour’ and ‘Passive Response’ had clearly developed. This is consistent with the finding that abnormal behaviour display may be caused by a variety factors (e.g., genetics, rearing conditions, housing and feeding [48]).

## 5. Conclusions

We conclude that under combined restrictions of social contact, social companionship, free movement and access to roughage horses display signs of suffering. The literature available on individual so called “basic needs”, does not allow us to isolate the effect of each of them, especially in the case of social companionship. However, the development of abnormal behaviour and passive coping strategies, can be considered signs of suffering, and these were displayed under separate restrictions in social contact, free movement and free access to roughage, as well as under combined restrictions of two or more of the proposed basic needs.

## Figures and Tables

**Figure 1 animals-11-01798-f001:**
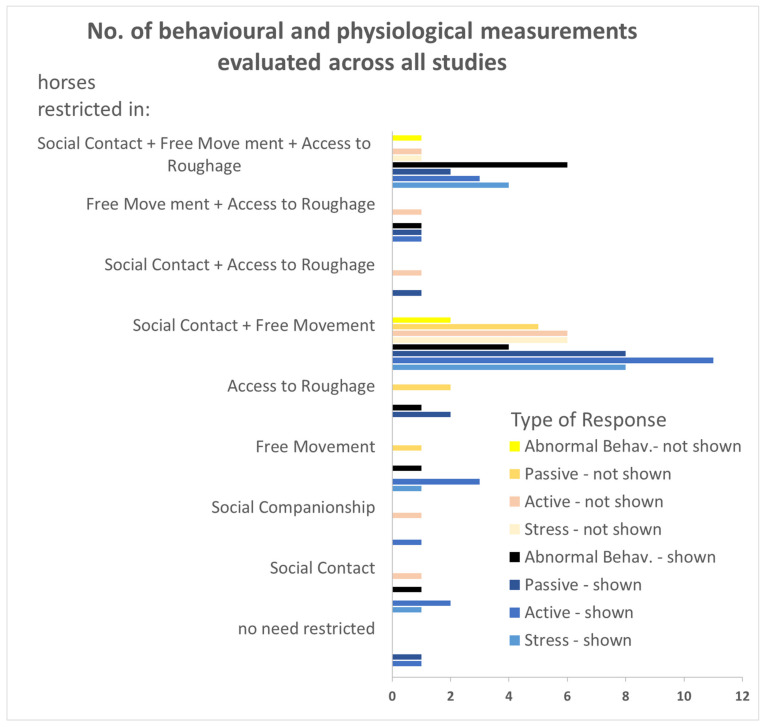
Number of reports of responses (dark colours, i.e., black and blue) versus number of reports of no response (light colours: skin and yellow) to restrictions in proposed “basic needs” described for response types across all studies. Abnormal Behaviour responses were significantly shown across all studies (Binomial Test: *p* = 0.02). Responses were also shown when social contact, free movement and access to roughage were mutually restricted (Binomial Test: *p* < 0.001). More detailed information on the responses, the test conditions and the horses included in the studies are given in Table 1 and in the Appendix A.

**Table 1 animals-11-01798-t001:** Literature on restrictions in “proposed basic needs” in horses. Background colours indicate studies on the restricted basic needs: grey = no restriction, light orange = social contact, light blue = social companionship, light yellow = movement, dark grey = feed, orange = social contact and feed, green = movement and social contact, yellow = movement, social contact and feed. The columns list the test condition or changes in test conditions, the response measurements and the horses’ responses that were observed. For detailed information on the horses and their management conditions please see Appendix A.

Reference	Restriction	Condition(s) Observed	Measurements	Response(s)
Hoffmann et al. 2012 [11]	No restriction	Group housing-no other conditions	Body condition score Behaviour: aggression Behaviour: social hierarchy Behaviour: synchronisation	Good Low Stable Good
Christensen et al. 2002 [36]	Social contact	Behaviour of young horses that had either been raised in group housing or single boxes were compared with behaviour when the horses were put out to pasture with other horses of the same age	Nearest neighbour	Horses from group housing sought proximity to former stable mates
			Behaviour: aggression	Higher in previously singly housed horses
			Behaviour: agonistic encounters (action and retreat response)	More subtle encounters in previously group housed horses
			Behaviour: social grooming	More frequent in previously singly housed horses
			Behaviour: play	More frequent in previously singly housed horses
Cooper et al. 2000 [33]	Social contact	Comparison between different types of single box housing: F: front top-half of the door open with a view of the stable courtyard FB: front half-door open and a similar half-door open at the back of the stable with a view to the surrounding fields B: back open only FS: front and one-side panel open with a view into the adjacent stable ALL 4: front, back and both sides open	Stereotypies: weaving	Most common prior to feeding in the morning and prior to putting out to pasture in the afternoon. Less weaving in the FS and All4 designs than the F design
			Stereotypies: repetitive nodding	FB, B, FS and All4: less nodding than in the F treatment
Hartmann 2010 [18]	Social contact	5 min social isolation from group housing (individually or in pairs)	Heart rate Behaviour: towards humans	No change No change
Nicol et al. 2005 [40]	Social contact	Comparison between barn and paddock weaned foals	Stress	Higher levels of stress in barn weaned foals
Christensen et al. 2011 [50]	Social companionship	Comparison between housing in unstable (changing) groups and stable (constant) groups	Behaviour: agonistic	More in unstable groups
			Behaviour: agonistic with contact	More in unstable groups
			Behaviour: greeting	More in unstable groups
			Behaviour: play	More variable in unstable groups
			Behaviour: agonistic (further behaviours)	No difference between housing groups
			Behaviour: affiliative (further behaviours)	No difference between housing groups
Chaplin and Gretgrix 2010 [39]	Movement	Same horses compared under Fully stabled (FS), Partially stabled (PS), Yard (Y), and Paddock (P) conditions	Activity: time spent active	More active on release from FS and PS housing
			Activity: time spent lying down	No change
Flauger and Krueger 2013 [34]	Movement	Different sizes of group paddock	Behaviour: aggressive	Decreased with increasing size of group paddock
			Behaviour: submissive	Decreased with increasing size of group paddock
Hoffmann et al. 2009 [20]	Movement	Provision of additional movement on pasture or in horse walker	Stress: faecal glucocorticoids	Decreased after movement
			Stress: heart rate variability	Decrease of sympathetic activity after movement
			Activity: movement	Increased
McGreevy et al. 1995a [45]	Movement	Length of time spent in single box housing	Stereotypic behaviour	Increased with time spent in individual boxes
Brinkmann et al. 2013 [51]	Feed	Feed restriction	Body condition score	Decreased
			Total bilirubin	Increased
			NEFA	Increased
			Total bilirubin and beta-hydroxyburyrat	Higher in males than in females
			Thyroxine concentrations	No change
Brinkmann et al. 2014 [52]	Feed	Metabolic adaptation to environmental conditions, comparing different quantities of feed and summer and winter temperatures	Body condition score	Reduced in winter
			Body mass	Reduced in winter
			Resting heart rate	Reduced in winter
			Metabolic rate	Reduced in winter
			Nocturnal hypothermia	Increased in winter
McGreevy et al. 1995b [46]	Feed	Stabled without straw bedding and with less feed than 6.8 kg forage/day	Abnormal behaviour	Increased
Jørgensen et al. 2011 [42]	Social contact & feed	Single turnout on paddocks versus group turnout. Feeding grass and roughage	Behaviour: passive	Increased when turned out singly
			Behaviour: passive	Reduced when fed with roughage or grass
			Behaviour: item exploration	No difference
Aurich et al. 2015 [23]	Movement & social contact	Group versus individual housing	Stress: salivary cortisol	No significant difference
Erber et al. 2013 [19]	Movement & social contact	Transfer from group housing to individual housing with initial riding	Stress: salivary cortisol	Increase after transfer
			Stress: heart rate	Increase during transfer
			Stress: heat rate variability	Decrease after transfer and after riding (= increase of sympathetic activity)
			Activity: locomotion	Decrease after transfer
			Stress: salivary cortisol	No change between housing conditions
			Stress: heart rate	No change between housing conditions
Fureix et al. 2012 [22]	Movement & social contact	Horses showing normal and horses showing withdrawn posture under conditions of no free movement and no free social contact	Stress: plasma cortisol	Low after work-further decrease with increased withdrawn posture
			Activity: body posture	Withdrawn posture 1–4 times every 30 min
			Activity: head, ear, eye movement	Reduced in withdrawn posture
			Activity: response to tactile stimuli	Reduced in withdrawn posture
			Activity: response to sudden approaching person	Reduced in withdrawn posture
			Activity: response to novel objects	Reduced in withdrawn posture
Harewood and McGowan 2005 [16]	Movement & social contact	Group versus individual housing	Behavioural scores	Higher in individual than in group housing
			Stress: heart rate	No difference
			Stress: salivary cortisol	No difference
			Diurnal rhythm heart rate and salivary cortisol	No diurnal rhythm under either condition
Heleski et al. 2002 [12]	Movement & social contact	Paddock-kept weanlings versus stable housed weanlings	Nearest neighbour	Paddock weaned foals stayed near conspecifics for longer
			Activity: grazing	Higher in paddock weaned foals
			Behaviours	Greater variety in paddock weaned foals
			Abnormal behaviour	Greater in stable weaned foals
			Stress: faecal glucocorticoid metabolites	No difference
Löckener et al. 2016 [10]	Movement & social contact	Living at pasture with social contact following single box housing	Behaviour: positive cognitive bias	Enhanced in horses on pasture with social contact
Niederhöfer 2009 [17]	Movement & social contact	Comparison between group housing, single box without paddock, and single box with paddock	Stress: faecal glucocorticoid metabolites	Lower in group housing
			Stress: heart rate variability	Lower in group housing
			Abnormal behaviour	Circling in horses in single boxes without paddock
Pell and McGreevy 1999 [44]	Movement & social contact	Stable housing compared to keeping at pasture	Abnormal behaviour	More frequent in stabled horses
Rivera et al. 2002 [15]	Movement & social contact	Stable housing versus keeping at pasture	Stress: heart rate	Lower in stabled horses
			Illness: gastric acidosis	More frequent in stabled horses
			Trainability: duration training procedure	Longer in stabled horses
			Trainability: duration habituation groundwork	Longer in stabled horses
			Trainability: head neck extension during training	Greater in stabled horses
			Behaviour: bucking and jumping	More frequent in stabled horses
			Stress: plasma cortisol	No difference
			Trainability: between mount and dismount	No effect of housing conditions
Ruet et al. 2019 [43]	Movement & social contact	Housing with window opening towards the external environment and straw bedding compared with housing with no window and non-straw bedding, different forage: grain feeding ratios and meal frequencies	Behaviour: aggression	Lower in housing with window and straw bedding
			Stereotypies: oral	Higher with grain feeding
			Stereotypies: oral	Number of meals per day had no effect
			Trainability: equitation and training	No difference
Sondergaard and Ladewig 2004 [35]	Movement & social contact	Effect of single versus group housing on training	Activity: restlessness before training	Greater in single housed horses
			Behaviour: biting, kicking during training	More frequent in horses in single housing
			Behaviour: defecation during training	More frequent in horses in single housing
			Trainability	Horses in group housing passed more training stages
Vitale et al. 2013 [53]	Movement & social contact	paddock turnout versus individual box housing versus fixed in a stock	Stress: heart rate variability	Decreased with reduced locomotion (= increased sympathetic activity)
Werhahn et al. 2011 [38]	Movement & social contact	No turnout compared to turnout	Behaviours: standing alert, aggression, occupation with equipment, occupation with bedding, dozing, sternal recumbency and lateral recumbency	More frequent in the horses with no turnout
			Activity: walking, standing/watching	More frequent in the horses with no turnout
			Trainability: willingness to perform	Enhanced in horses with turnout
			Trainability: duration of training	Shorter in horses with turnout
			Locomotion	No difference
Werhahn et al. 2012 [21]	Movement & social contact	Single box housing, individual turnout, group turnout	Stress: heart rate variability measures SDNN, RMSSD and LF/HF	Higher sympathetic activity when horses were stabled in single boxes
			Behaviour: lying down	Longer when horses had group turnout
			Trainability: willingness to perform	Slightly better when the horses had turnout
			Behaviour: standing alert, dozing, eating, occupation	No change
			Locomotion	No change
Wille 2010 [24]	Movement & social contact	Open barn housing, individual box housing, tied up in stalls	Stress: faecal glucocorticoid metabolites	Lower in open barn system
			Behaviour: standing	Longer when tied in stalls
			Behaviour: lying on the chest	Longer in open barn system or individual boxes
			Behaviour: lying on the side	Longer in open barn system
			Locomotion	More in open barn system
			Behaviour: food consumption	No difference
Yarnell et al. 2015 [13]	Movement & social contact	Single housing with no contact (SHNC), group housing with full contact (GHFC), paired housing with full contact (PHFC)	Stress: faecal glucocorticoid metabolites	Higher in SHNC
			Stress: eye temperature	Lower in GHFC
			Behaviour: standing	Lower in GHFC
			Behaviour: active and social negative behaviours	Higher in GHFC and PHFC
			Trainability: handling	More difficult with SHNC horses
Lesimple et al. 2020 [54]	Movement & feed	Change from single box with no paddock to housing with turnout and ad lib hay	Behaviour: vigilance, excitement and locomotion	Decreased with turnout and ad lib hay
			Behaviour: feeding with ears laid back	Decreased with turnout and ad lib hay
			Stereotypies	Decreased with turnout and ad lib hay
			Blood: oxytocin	Increased with turnout and ad lib hay
			Blood: cell counts, serotonin	No change
Bachmann et al. 2003 [48]	Movement, social contact & feed	Restricting feed and daily pasture	Stereotypies: crib-biting, weaving and box-walking	Increased
Mal et al. 1991 [14]	Movement, social contact & feed	Horses of different temperaments; housing in isolation (ISS), at pasture (P), in individual boxes with social contact (C)	Behaviour: time eating grain, grain-eating bouts	More in horses of medium and highly reactive temperaments in isolation (ISS)
			Behaviour: forage-eating bouts	Longer in calm horses at pasture
			Activity: distance travelled, time spent trotting, number of trotting bouts, number of standing bouts, number of total activity bouts	More in isolation horses
			Activity: duration standing	Less in isolation horses
			Triiodothyronine	Highest in isolation horses
Marr et al. 2020 [25]	Movement, social contact & feed	Change from group housing to individual housing, and initial training	Stress: faecal glucocorticoid metabolites	Increased after change from group to individual housing after 24 h, 48 h, and 1 week. Increased after 24 h, 48 h, and 2 months of initial training
			Behaviour: motor laterality	Left shift for 1 week after change from group to individual housing, and after 2 months of initial training
			Behaviour: sensory laterality	Left shift 24 h after change from group to individual housing, and (not significantly) 24 h after initial training
			Stress: Immunoglobulin A	Decreased (not significantly) after change from group to individual housing, and 24 h after initial training
Redbo et al. 1998 [55]	Movement, social contact & Feed	Thoroughbreds compared with trotters	Behaviour: wood-chewing	No difference
			Stereotypies	More in thoroughbreds
Stomp et al. 2021 [26]	Movement, social contact & feed	Hemispheric activity in horses in individual housing compared with that in horses kept at pasture	Electroencephalogram (EEG): bilateral predominance of theta waves	Increased in pasture kept horses.
			EEG: bilateral predominance of beta waves	Increased in horses in individual housing
			EEG: hemispheric laterality: bilateral and Left-Hemispheric theta activity	Increased in pasture kept horses
			EEG: hemispheric laterality: bilateral or Right-Hemispheric high production of gamma waves	Increased in horses in individual housing
			Stereotypies	More common in horses in individual housing
			Behaviour: ear position while feeding	More common in horses in individual housing
			Behaviour: human approach-tests	Pasture kept horses more positive
Visser et al. 2008 [41]	Movement, social contact & feed	Housing in individual boxes versus housing in pairs	Stress: CRF challenge test-cortisol response and ACTH response	Lower in individually housed horses
			Behaviour: neighing, pawing, nibbling, snorting	More frequent in individually housed horses
			Stereotypies	More frequent in individually housed horses
			Activity: novel object test	No difference
Waters et al. 2002 [8]	Movement, social contact & feed	Weaning in a stable, a barn, on a paddock, and at grass	Abnormal behaviour	More frequent after weaning in barns or stables
			Stereotypic behaviour: wood chewing	More frequent after weaning in barns or stables

## Data Availability

All data are provided in the manuscript and the Appendix A.

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
