# Peer review of "Basic Needs in Horses?—A Literature Review"

_animals, 2021, doi:10.3390/ani11061798_

Round 1

Reviewer 1 Report

Dear  Authors, The manuscript:" Basic need in horses-? A literature review" summarizes research that described disturbances in basic needs in horses. I find this work interesting and valuable. However, while the results and discussion are very well written the Introduction and huge Table 1with material and methods let me a bit confused.  I suggest, to rewrite the Introduction. At the moment is really Table 1 explanation, which should be in material and methods. 

Below my detailed comments

ln 51: delete and can be

ln 53: do you established this in this manuscript

Ln 55: delete therefore, 

ln 56: delete whether, 

ln 73-74; the word "here" suggests that is for purpose of the study, so this should be in material and methods section

ln 85: delete "furtheremore"

ln 83: same like ln -73-74.

Ln 123-134: this should be considered in material and methods. 

Author Response

Dear Referee, thank you very much for your valuable suggestions. They were most helpful in   enhancing the readability and the applicability of our manuscript „Basic needs in horses? – a literature review“

We have rewritten and shortened the introduction, and highlighted the importance of the topic for animal welfare analysis in equine management. Aspects concerning the content of table 1 have been  moved to the Methods section.

In light of the suggestions of all the referees and the academic editor we restructured Table 1. We feel the present version is easier to read, and more helpful in terms of providing a good overview of the management restrictions, management conditions, measurements and responses of the horses described in the literature. Information on the structure of the table is now given in the table caption.

A language and spell check has been done by a native speaker for the text, the tables and the supplementary material.

All your minor suggestions were implemented.

Best regards on behalf of all the authors,

Konstanze Krüger

Reviewer 2 Report

 I did not find that this paper added much to what is already known about the impacts of restrictions placed on horses in managed situations. It was however really good to see statistical analysis applied to the findings of research in this area derived from multiple studies. There were multiple places in this report that were difficult to follow - although this can be relatively easily fixed with some rephrasing and rewording.   In some sections there was a proliferation of the use of 'we'.   I found this really distracting from the fact that a scientific study had been conducted.   There were several items suddenly discussed at the end of the discussion with minimal introduction earlier in the paper.  I have made many comments on the actual paper (fully intended to be constructive). I hope that these are of use. 

I wonder if a more applied focus could be added throughout this paper, to enable it to add to what is already known, at present it does feel a little as if an opportunity has been missed.

Author Response

Dear referee 2

Thank you so much for your thoughts and suggestions for our manuscript „Basic needs in horses? – a literature review. Your points are well taken.

We have rewritten and shortened the introduction, and highlighted the importance of the topic for animal welfare analysis in equine management. Aspects concerning the content of table 1 have been  moved to the Methods section.

In light of the suggestions of all the referees and the academic editor we restructured Table 1. We feel the present version is easier to read, and more helpful in terms of providing a good overview of the management restrictions, management conditions, measurements and responses of the horses described in the literature. Information on the structure of the table is now given in the table caption.

A language and spell check has been done by a native speaker for the text, the tables and the supplementary material.

We changed the unclear wording as you suggested throughout the manuscript.

The keyword „depressive like state“ has been removed, as this is only a minor aspect.

Line 51: The statement „As a general assumption,…“ is based on the literature given at the end oft he sentence.

Thank you for drawing our attention to the colours of figure 1. We changed them slightly, so that important aspects are clearly visible in a black and white version as well.

Best regards on behave of all the authors,

Konstanze Krüger